# Combination of Graphene Oxide and Rhizobium Improved Soybean Tolerance in Saline-Alkali Stress

Xiaohong Fu [1,†], Dahong Bian [2,†], Xuyang Gu [1], Jinfeng Cao [3] and Jianfeng Liu [1,*]

1　School of Life Sciences, Institute of Life Science and Green Development, Hebei University, Baoding 071002, China; xhxh119_01@163.com (X.F.); guxu880_98@163.com (X.G.)
2　State Key Laboratory of North China Crop Improvement and Regulation, College of Agronomy, Hebei Agricultural University, Baoding 071001, China; cracy15@163.com
3　Hebei Key Laboratory of Crop Salt-Alkali Stress Tolerance Evaluation and Genetic Improvement, Cangzhou 061001, China; caojinfeng2003@163.com
*　Correspondence: jianfengliu@hbu.edu.cn
†　These authors contributed equally to this work.

**Abstract:** Soybean (*Glycine max* L.) is the most important crop plant in the world. Soil saline-alkali seriously inhibits soybean yield and quality. This study aims to investigate the impact of graphene oxide (GO) and *Rhizobium* (Rh) on the expression of soybean-related genes as well as the growth and yield under saline-alkali stress. The results show that GO + Rh-treated increased the number of root nodules by 5.43 times compared with the control (Ctrl), the total nitrogen content and root system parameters of plants were also significantly improved. GO + Rh-treated reduced the $Na^+/K^+$ ratio and the osmotic substances, while the activities of antioxidant enzymes SOD, POD, CAT and APX in GO + Rh-treated soybean plants increased significantly by 69.18%, 69%, 75.64% and 48.38% compared with the control plants. The REC, MDA and $H_2O_2$ content decreased significantly by 46.73%, 42.80% and 43.53%. In addition, GA3 content, among all related saline-alkali hormones, was increased by 100.20% compared with the Ctrl. The expression level of *GmGBP1*, a key gene for GA3 synthesis, at most increased 6.42 times compared to the Ctrl. The results further reveal that GO + Rh-treated obviously improves the yield traits of soybean plants, which confirms that GO + Rh-treated could be effective in enhancing soybean tolerance to saline-alkali stress. Our findings provide a new strategy for improving the saline-alkali tolerance of soybean, as well as a new perspective for exploiting and utilizing large-area saline-alkali soil.

**Keywords:** soybean; saline-alkali stress; graphene oxide; *rhizobium*; antioxidant enzyme system; RT-qPCR

## 1. Introduction

Soybean (*Gilcine max* L.) as an important food and oil crop is rich in protein and fat [1]. Due to various extreme environments, the yield and quality of soybeans are seriously affected, among which soil saline-alkali seriously limits soybean growth and yield [2]. Studies have shown that saline-alkali stress could induce stomatal closure of soybean leaves, thus reducing photosynthesis and transpiration rate [3]. The high salt environment slows down soybean root growth, which reduced the absorption of the nutrients from soil [4]. High soil saline-alkali destroys plant cell homeostasis and physiological and biochemical processes, and excessive $Na^+$ produces toxic effects and destroys the ion balance of soybean, leading to osmotic stress and water deficit [5]. The accumulation of harmful ions induces plants to produce a large number of reactive oxygen species, which damages the cell membrane and destroys the homeostasis of plants, eventually reducing the photosynthetic rate and inhibiting plant growth [6]. Moreover, the root growth is hindered in high concentration saline-alkali environments, which has a destructive impact on the metabolism and transportation of plants [7]. Soybeans exposed to alkali stress will also

suffer the same osmotic stress and oxidative damage as salt stress. Especially, the high pH value caused by soil alkalization seriously inhibits ions absorption by plants. To overcome the toxicity caused by saline-alkali stress and eliminate excess ROS, plants balance the osmotic potential and maintain normal growth and development through the repair of the antioxidant enzyme system and ion transport while the self-regulation of plants is very limited [8]. At present, the main strategies to improve saline-alkali tolerance of soybean are gene editing or transgenic methods. However, molecular breeding for soybean varieties to improve saline-alkali resistance still has some insurmountable problems, such as the lack of genetic resources, and time-consuming and inconvenient inspection. Therefore, a kind of green, high-efficiency, cost-saving strategy is a requisite to facilitate plant growth.

Graphene oxide (GO), with more functional groups, is the oxidized state of graphene. It has better water solubility and good reactive oxygen removal capacity, which is widely used in various fields, including pharmaceutical carrying, biological treatment, and agricultural production. Particularly application in the agricultural field is gradually extensive for improving plant resistance to adverse responses [9]. GO application could also enhance physiological indices and enzyme activity resulting in increasing the resistance of crops to abiotic stress [10]. Studies have revealed that GO applies to maize's (*Zea mays* L.) growth and development impacts [11]. Therefore, GO has a vital function in the growth and development of plants under stress.

GO concentration has a different impact on plant growth, and excessive concentration might cause toxic effects. For seed germination, a low dose of GO (0.1–10 mg/L) could significantly improve the situation of wheat germination, and prolong the growth period of rose [12]. An amount of 25 to 100 mg/L GO inhibits rape (*Brassica napus* L.) root growth, but significantly increases the content of ABA [13]. Overall, GO could promote seed germination and plant growth at a suitable concentration. Especially, GO improves the ability of plants to resist abiotic stress.

Most leguminous plants could be symbiotic for nitrogen fixation with a variety of bacteria, collectively known as *Rhizobium* [14]. Studies also show that the inoculation of soybean with *rhizobium* could significantly increase nitrogen nutrition, leaf area, and chlorophyll content, and reduce root and leaf relative electrolytic leakage (REL), further improving the ability to resist salt stress [15]. Moreover, the combination of *Rhizobium* and nano iron oxide can also improve the capability of *alfalfa* to resist soil cadmium pollution [16] and the combination of exogenous gibberellin. Additionally, *rhizobium* could improve the nodule numbers and nitrogen absorption ability of *alfalfa*, and promote growth [17]. *Rhizobium* has a good application prospect in interacting with other substances to resist adverse environmental impacts. However, there are few reports about the effects of the combination of GO and *rhizobium* on the saline-alkaline tolerance, and nitrogen fixation of the plants.

Studies reveal that GO could promote the growth of *rhizobium*, and increase the nodule numbers to improve the nitrogen fixation ability of plants [18]. In our findings, we demonstrate that GO and Rh combination constructions can improve tolerance of saline-alkali stress in soybean plants, which could significantly increase soybean phenotypic performances and indexes, defense enzymes activity, nitrogen fixation ability, hormone levels, and hormone-related genes expression level under saline-alkali. Furthermore, we find that GO and Rh combination also has an active effect on the yield traits of soybean. Our results provide new insights for enhancing saline-alkali tolerance of soybean plants by application of GO + Rh biological agents.

## 2. Material and Methods

### 2.1. Experimental Materials

Soybean seeds were provided by the Cangzhou Academy of Agricultural Sciences, and the variety was Cangdou 1438. The concentration of graphene oxide (GO, Suzhou Tanfeng Company, Suzhou, China.) was 30 µg/mL. The effective viable count of *Bradyrhizobium japonicum* is greater than $3 \times 10^9$ (Rh, Shanghai biology Company, Shanghai, China).

The germinating seeds were set in 500 mL hydroponic bottles filled with 1/2 Hoagland solution and 80 mmol/L saline-alkali solutions, and grown in a growth chamber (28/20 °C, 75% humidity) and provided with a photoperiod for 16 h. We soaked the germinated seeds in *Bradyrhizobium japonicum* solution for 10 min, and inoculate them to the roots for 3 mL when transplanting them. The plants were classified into five treatments when they grew to 4–6 true leaves, each for 15 bottles. The detailed treatment of the five groups is shown in Table 1.

**Table 1.** Treatment methods of hydroponics.

| Abbreviation | Mixture Components |
| --- | --- |
| No treatment | Hoagland solution |
| Ctrl | Hoagland solution+ saline-alkali solution |
| GO | Hoagland solution + saline-alkali solution + 100 mL GO |
| Rh | Hoagland solution+ saline-alkali solution + 3 mL Rh |
| GO + Rh | Hoagland solution+ saline-alkali solution + 100 mL GO + 3 mL Rh |

Pot-culture: Soybean was planted in flowerpots and placed in the garden of College of Life Sciences, Hebei University. Saline-alkali treatment began at the full flowering stage of soybean, and each treatment is shown in Table 2.

**Table 2.** Treatment details of pot culture.

| Abbreviation | Mixture Components |
| --- | --- |
| No treatment | normal watering |
| Ctrl | saline-alkali solution |
| GO + Rh | saline-alkali solution + 100 mL GO + 3 mL Rh |

The samples of soybeans were collected at the seedling stage and mature stage, treated with liquid nitrogen and stored in the refrigerator at −80 °C. Hydroponic culture treatment and pot culture for soybean plants were designed as shown in Figure 1.

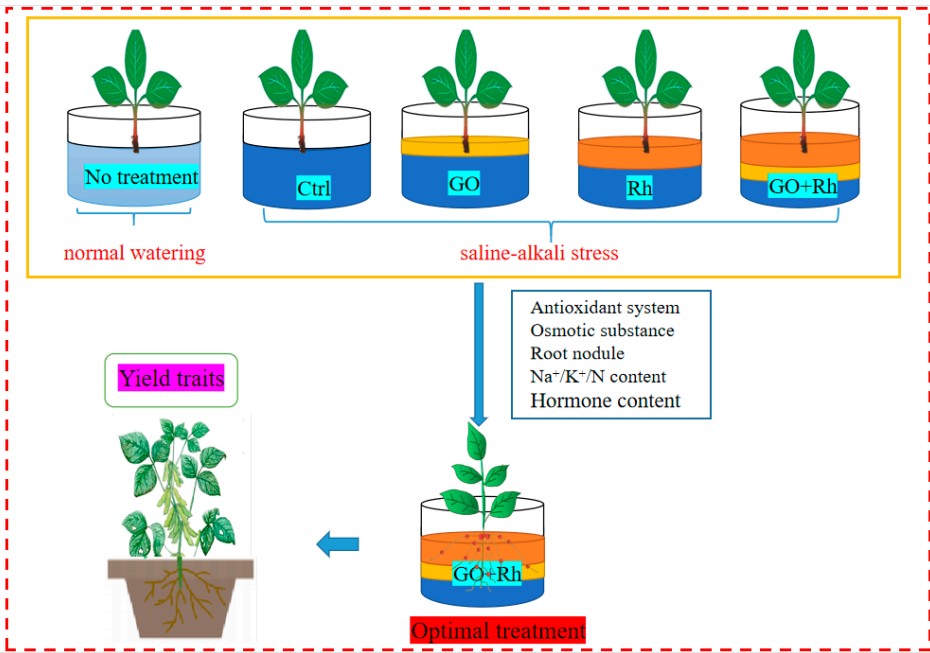

**Figure 1.** Schematic diagram of our experimental method.

### 2.2. GO Characteristics

The GO solution was first observed with the transmission electron microscope (TEM, Tecnai-G20, FEI, USA), environmental scanning electron microscopy (SEM, MFP-3D, Oxfod, UK), Raman spectroscopy (InVia2000, Renishaw, London, UK), and Fourier transform infrared spectroscopy (FITR, Nicolet-460, Thermo Fisher, CT, USA). Further, the GO solution was dispersed in distilled water by ultrasound (Kunshan Shumei Ultrasonic Instruments Co., Ltd., Suzhou, China) with a 2 h ultrasonic duration and >300 W ultrasonic power, and the temperature was controlled below 20 °C during ultrasonication.

### 2.3. Determination of Morphological Index

Fresh weight (FW) of the plant tissue was weighed immediately, then dry weight (DW) was measured after blanching at 105 °C for 15 min and drying to constant weight at 75 °C by an oven. The absolute water content was calculated: (FW-DW)/DW $\times$ 100%. The root morphological parameters were determined using a root system scanner (Seiko Epson Corp., Tokyo, Japan). The root total length, root volume, and root surface area were measured by WinRHIZO 4.0b software.

### 2.4. Determination of Physiological Indexes Related to Saline-Alkali Tolerance

The accumulation of hydrogen peroxide ($H_2O_2$) usually was determined by the diaminobenzidine (DAB) staining method [19]. The soybean seedling's leaves were fully soaked in DAB staining solution at 25 °C for 2 h, after extracting the DAB solution and adding ethanol (70%). The soybean leaves were boiled to observe the leaves' color [20]. $H_2O_2$ contents were measured by a kit (No. A064-1-1, Nanjing Jiancheng Bioengineering Co., Ltd., Nanjing, China). The 0.1 g soybean leaves were pestled, and then 0.9 mL physiological saline was added to gain a supernatant. Then, it was centrifuged (3500 rpm, 4 °C) for 10 min. The preparation solution was determined at 420 nm.

The relative electrical conductivity (REC) was measured by a conductivity meter (DDBJ-350) [19]. Malondialdehyde (MDA) was detected by the method with thiobarbituric acid [20] (No. A003-1-1). The free proline (Pro) was analyzed with a kit (No. A107-1-1) by colorimetric method. All the above kits are from Nanjing Jiancheng Bioengineering Co., Ltd., Nanjing, China.

Determination of $Na^+$ and $K^+$ content in tissue by Atomic Flame Photometer (FP6410, Shanghai Yidian, Shanghai, China) [21].

### 2.5. Determination of Enzymatic Activities

Reagent kits were used to determine superoxide dismutase activities (SOD, No. A001-1-1), peroxidase activities (POD, No. A048-3-1), catalase activities (CAT, No. A007-1-1), and ascorbate peroxidase activities (APX, No. A107-1-1). All the above kits are from Nanjing Jiancheng Bioengineering Co., Ltd., China.

### 2.6. Determination of Total Nitrogen Content

Determination of total nitrogen content with the Kjeldahl method [22]. Using a KDN-520 (Lü bo). The total nitrogen is calculated as follows:

$$N \ (mg/g) = [(V_1 - V_2) \times M \times 0.028]/G \times 100.$$

Measure of leghaemoglobin: Take a proper amount of root nodules (W) and grind them into homogenate in phosphoric acid buffer at 4 °C, the amount of phosphoric acid buffer is about 4 times the volume of nodules, centrifuge at 4 °C for 15 min (V), discard the sediment, and continue to collect the supernatant. Centrifuge at 4 °C for 20 min, and then put the supernatant in a spectrophotometer. Colorimetric determination of absorbance at 540 nm, mark it as A [23]. The calculation formula is as follows:

$$C_{Lb} \ (g/L) = A \times 367.7 \ (g/L)$$

$$\text{Leghemoglobin content (mg g) = } C_{Lb} \times V/W$$

The activity of nitrogenase was determined by the acetylene reduction method [23]. 0.3 g nodules (m) were set into a serum bottle, inject acetylene with an air ratio of 1:100 into the bottle, and incubate in a 28 °C incubator in the dark for 2 h (t). Use a gas mass spectrometer (Thermo Scientific, Trace GC Ultra, USA) to measure the ethylene content, and record it as X. The calculation formula is as follows:

$$\text{Nitrogenase activity (nmol/mg·h) = X/m} \times t$$

### 2.7. Determination of Hormone Levels

The related saline-alkali hormones, such as ABA, IAA, JA, and GA3 were detected by Zoonbio Biotechnology. The product of samples was analyzed with the ZORBAX SB-C18 column (Agilent Technologies) of the high-performance liquid chromatography-tandem mass spectrometry (HPLC-MS/MS) method. The solvents for the mobile phase were ultrapure water and 0.1% methanoic acid, methanol, and 0.1% methanoic acid, and injected volumes were 2.5 μL. Mass spectrometer (MS) treatment was completed at a voltage of 4000 V. The pressure of the nebulizer, air curtain, and aux gas was 60, 20, and 75 psi, respectively, and the temperature of atomizing was set at 350 °C.

### 2.8. Determination of Gene Expression

Samples were collected at 0 h and 7 h after treatment, respectively, and RNA extraction was used by the RNA Kit (DNase I) (CW2598S, CWBIO). The primers of qRT-PCR were made by Primer-BLAST software. The soybean actin gene was taken as the internal reference gene, in which three biological repeats were set up and repeated three times. The reverse transcription kit is a 2 × Fast Super EvaGreen qPCR Master Mix kit. The temperature and time were set as 95 °C for 5 min, followed by 45 cycles of 95 °C for 15 s and 57 °C for 60 s. The expression levels of genes were calculated according to the $2^{-\Delta\Delta Ct}$ method. All primer sequences were shown in Supplementary Table S1.

### 2.9. Statistical Analysis

The indexes and gene expressions are displayed as means ± standard deviations (SD) basis on three independent experiments. All results were using Origin software (version 8.5) by a one-sample t-test. The GO, Rh, GO + Rh-treated samples were compared to the Ctrl except for no treatment. The $p \leq 0.05$ or $p \leq 0.01$ were considered significant. A principal component analysis (PCA) was used by OriginPRO (version 2021) to test the relationship between each sample and the variable.

## 3. Results

### 3.1. GO Characteristics

First, we obtained the surface characteristics of GO. Through the SEM technology, it was found that GO had a fold structure and presented a stacking state (Figure 2A). In addition, the TEM image showed that GO contained a multi-layer fold structure (Figure 2B). Using FITR technology, it was found that GO contained various oxygen-containing functional groups such as –OH, C–O, C=O, –COOH, and –O (Figure 1C). Further, as shown in the Raman spectrum, we detected that GO had two main characteristic peaks located in the D band of about 1349 cm$^{-1}$ and the G band of 1590 cm$^{-1}$, respectively (Figure 2D).

### 3.2. GO + Rh Enhanced Saline-Alkali Tolerance of Soybean Seedlings

The different treatments of Ctrl, GO, Rh, and GO + Rh were carried out to further analyze their effects on the soybean seedlings' growth under the saline-alkali stress. As expected, the Ctrl had significantly saline-alkali-sensitive symptoms with wilting leaves; whereas the GO-treated and Rh-treated soybeans' growth performance resulted in slightly wilted leaves. However, the soybean leaves of GO + Rh-treated showed no obvious changes

under saline-alkali stress, similar to the no-treatment soybeans performance (Figure 3A). Meanwhile, saline-alkali stress could reduce the physiological indicators such as fresh weight and water content of soybean plants, the GO + Rh-treated soybeans showed a better effect than the GO and Rh-treated soybeans. The GO + Rh-treated, GO-treated, and Rh-treated fresh weight of shoot and root and absolute water content were obviously increased by 153.57%, 92.06%, and 71.00% compared with those of the Ctrl soybeans (Figure 3B,C).

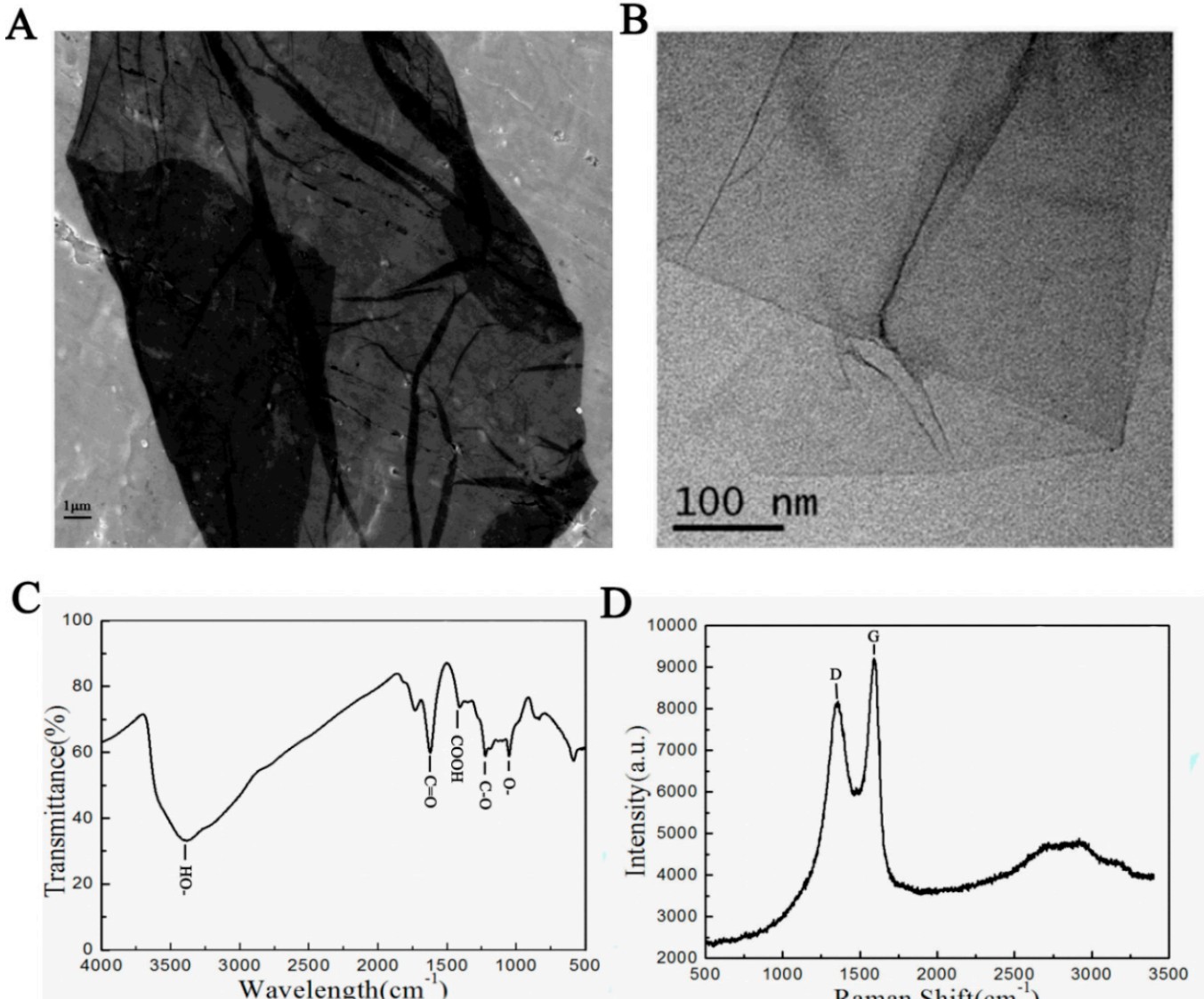

**Figure 2.** Observation of GO characteristics. (**A**) SEM image of GO. (**B**) TEM image of GO. (**C**) FTIR spectrum of GO. (**D**) Raman spectrum of GO.

Further, $K^+$ high uptake and $Na^+$ reduction is the important strategy for soybean plants during saline-alkali stress. Our results revealed that $K^+$ contents in the leaf were notably decreased under saline-alkali stress, but were improved with GO or Rh and GO + Rh-treated. The $K^+$ content of GO + Rh-treated performed best and increased by 23.05%, significantly compared to the Ctrl. $Na^+$ content in the leaf was obviously increased under saline-alkali stress resulting in an $Na^+/K^+$ ratio increased under saline-alkali stress. While $Na^+$ content was significantly decreased by 47.45% with the GO + Rh-treated compared to the Ctrl (Figure 3D,E). We also found that the $Na^+$ content in the hydroponic solution of GO + Rh-treated soybean was significantly higher than that of the Ctrl. This indicated that the GO + Rh-treated absorbed less $Na^+$ in the hydroponic solution, thus ensuring the ion

balance in vivo (Table 3). Our results revealed that GO + Rh might regulate positively in the growth development of soybean plants subjected to saline-alkali stress.

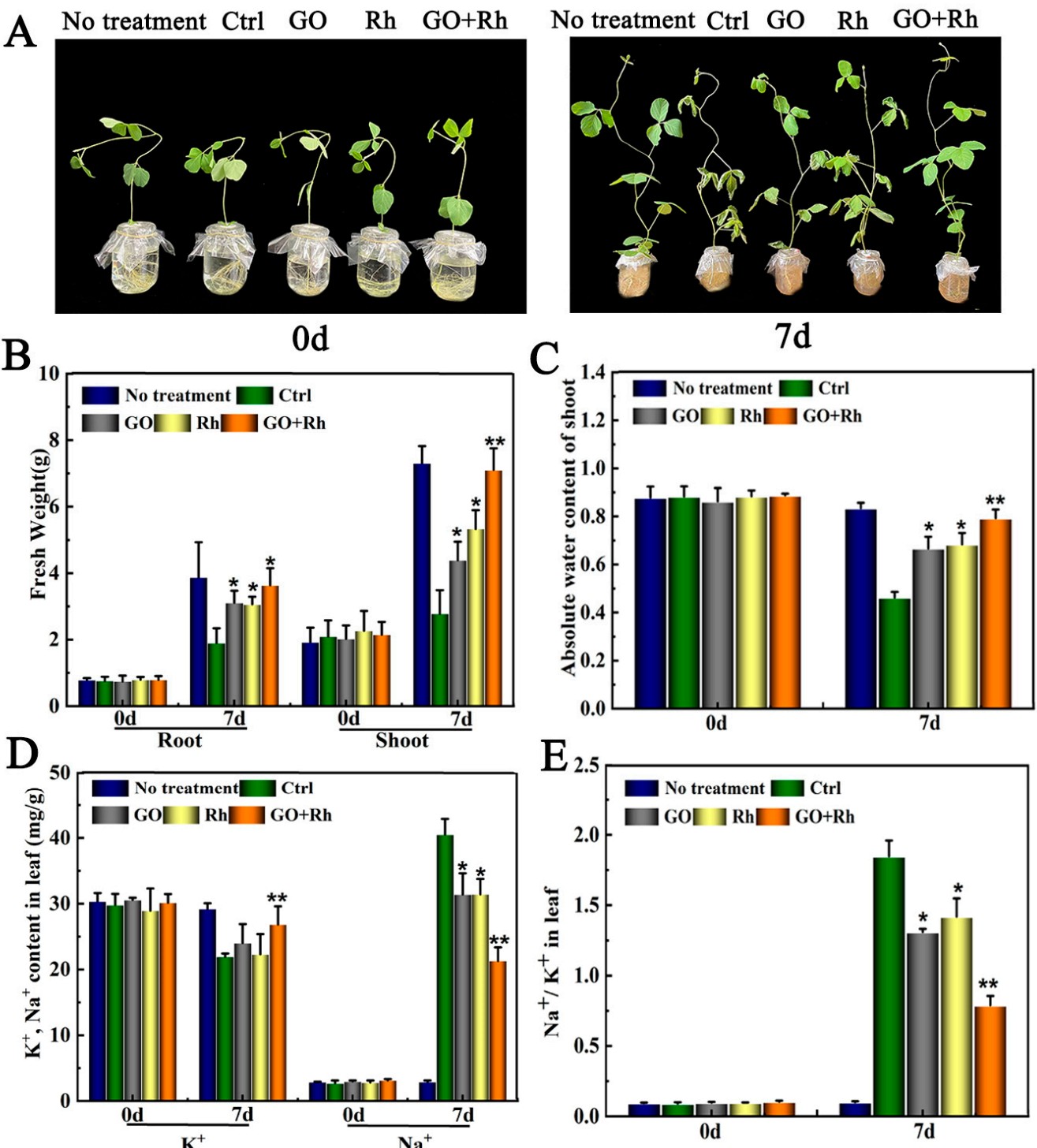

**Figure 3.** Effects of different treatments on growth performance of soybean seedlings under saline-alkali stress. (**A**) Phenotype of soybean at seeding stage. (**B**) Soybean fresh weight. (**C**) The absolute water of shoot. (**D**) The $Na^+$, $K^+$ content of leaf. (**E**) Leaf $Na^+/K^+$. Data showing the means $\pm$ standard deviation (SD). ($n$ = 3) (** $p \leq 0.01$,* $p \leq 0.05$).

**Table 3.** Content of Na$^+$ in water culture solution.

| Time | Treatment | Na$^+$ Content (mg/g) |
|---|---|---|
| 0 d | No treatment | 6.40 ± 0.68 |
| | Ctrl | 6.35 ± 0.82 |
| | GO | 6.44 ± 0.54 |
| | Rh | 6.51 ± 0.28 |
| | GO + Rh | 6.26 ± 0.46 |
| 7 d | No treatment | 8.46 ± 0.92 |
| | Ctrl | 219.08 ± 8.52 |
| | GO | 225.48 ± 5.73 |
| | Rh | 221.59 ± 4.79 |
| | GO + Rh | 237.87 ± 6.56 * |

Data showing the means ± standard deviation (SD). (*n* = 3) (* $p \leq 0.05$).

### 3.3. GO + Rh Promoted N Uptake and Improved Soybean Seedlings Root Parameters

Developing a strong root system is vital for soybean plants to promote absorbing water, nutrition, inorganic salts, and the transportation of nitrogen nutrition. We found that the root system of GO, Rh, and GO + Rh-treated soybeans were denser root systems than the Ctrl. All of theGO + Rh-treated acted not much differently in the growth performance compared to the no-treatment soybeans with optimum root systems under saline-alkali stress. In detail, the root total length, surface area, diameter, and volume of GO + Rh-treated were higher than the Ctrl by 57.19%, 54.99%, 20.54%, and 52.87%, which showed the best result compared to GO, Rh alone. (Figure 4A; Table 4). Further, the number of nodules of roots with GO + Rh-treated roots increased by 5.43 times, compared with the Ctrl roots (Figure 4B,C). The increase in the number of root nodules also promoted the improvement of the nitrogen fixation ability of roots, ensuring the supply of nitrogen nutrition (Figure 4D).

**Table 4.** Soybean root system parameters under saline-alkali stress.

| Time | Treatment | Root Length (m) | Root Surface (cm$^2$) | Root Diameter (mm) | Root Volume (cm$^3$) |
|---|---|---|---|---|---|
| 0 d | No treatment | 4.45 ± 0.96 | 55.23 ± 12.77 | 4.0 ± 0.18 | 0.57 ± 0.13 |
| | Ctrl | 4.43 ± 0.82 | 52.97 ± 5.13 | 3.97 ± 0.38 | 0.50 ± 1.0 |
| | GO | 4.52 ± 0.43 | 53.53 ± 3.34 | 3.76 ± 0.36 | 0.51 ± 0.06 |
| | Rh | 4.41 ± 0.11 | 54.37 ± 10.82 | 4.04 ± 0.11 | 0.56 ± 0.11 |
| | GO + Rh | 4.51 ± 0.27 | 56.72 ± 4.33 | 4.02 ± 0.40 | 0.57 ± 0.09 |
| 7 d | No treatment | 23.69 ± 0.44 | 278.46 ± 5.54 | 5.50 ± 0.48 | 2.61 ± 0.63 |
| | Ctrl | 14.53 ± 0.56 | 178.23 ± 9.57 | 4.01 ± 0.12 | 1.74 ± 0.13 |
| | GO | 18.56 ± 3.21 | 235.50 ± 38.71 | 4.20 ± 0.20 | 2.38 ± 0.37 |
| | Rh | 21.25 ± 0.44 * | 261.49 ± 11.19 ** | 4.36 ± 0.40 | 2.56 ± 0.19 * |
| | GO + Rh | 22.84 ± 3.24 * | 276.24 ± 28.61 ** | 4.83 ± 0.13 ** | 2.66 ± 0.16 * |

Data showing the means ± standard deviation (SD). (*n* = 3) (** $p \leq 0.01$,* $p \leq 0.05$).

To sum up, GO + Rh could promote root growth, nodulation and increase total nitrogen content to resist saline-alkali stress, and the effect was much higher than when GO, Rh was applied alone.

### 3.4. GO + Rh Increased Antioxidant Enzyme System Activity and Decreased Hydrogen Peroxide Content

The accumulation of hydrogen oxide in cells leads to oxidative stress. Catalase plays a key role in the antioxidant activity of the defense system. In this study, GO-treated, Rh-treated, and GO + Rh-treated all increased antioxidant enzymes SOD, POD, CAT, and APX activities in soybean with saline-alkali stress. It should be mentioned that GO + Rh-treated increased by 69.18%, 69%, 75.64%, and 48.38% more than the Ctrl,

respectively (Figure 5A–D). This positive effect was approximately double that of GO, Rh alone.

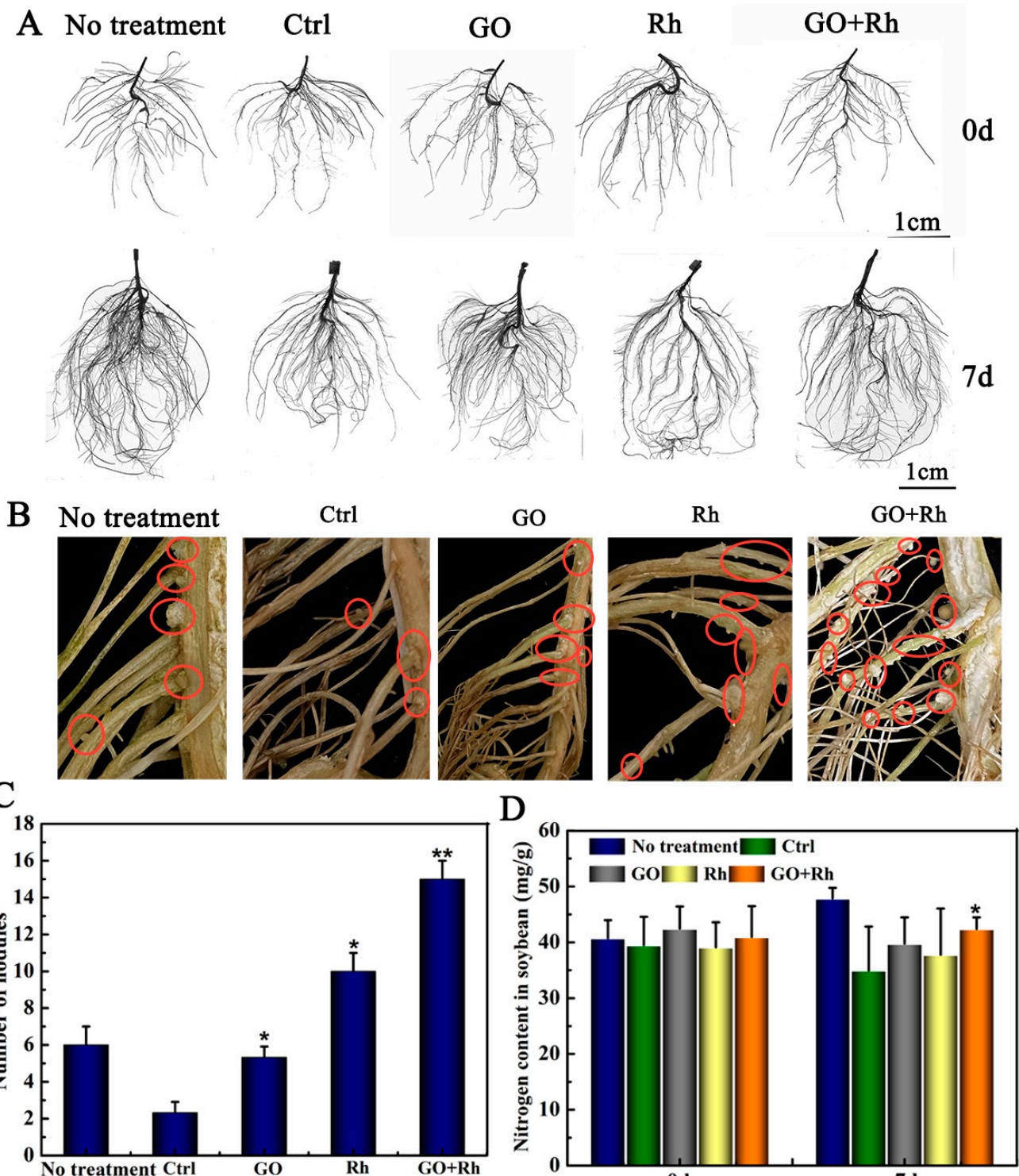

**Figure 4.** Different treatments effect on root system and nodule number of soybean under saline-alkali stress. (**A**) soybean root systems images scanned. (**B**) Phenotypic diagram of nodule number. (**C**) Number of nodules. (**D**) Nitrogen content in soybean. Data showing the means $\pm$ standard deviation (SD). ($n$ = 3) (** $p \leq 0.01$, * $p \leq 0.05$). Red circles are marked as the location of nodules in (**B**).

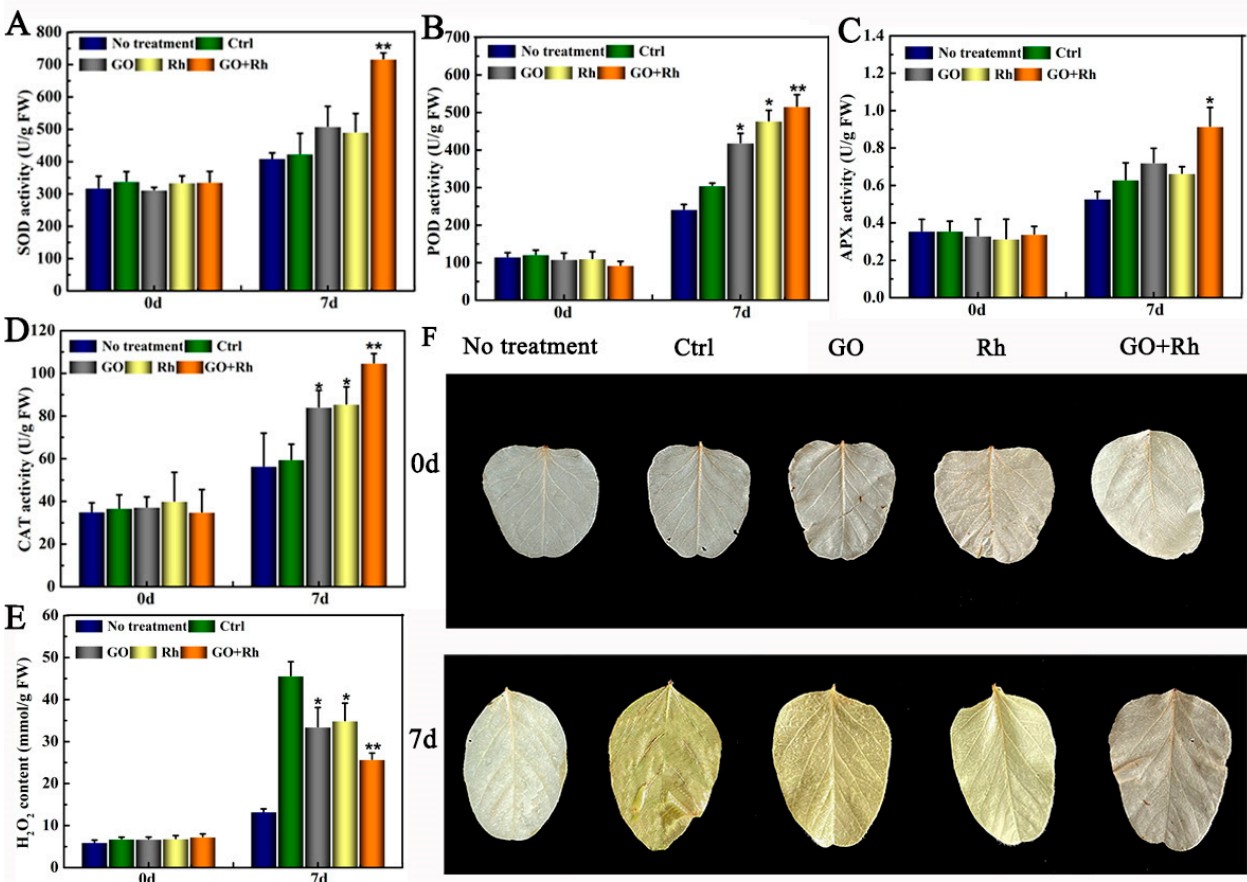

**Figure 5.** Different treatments' effect on antioxidant enzyme activity and $H_2O_2$ content of Soybean under saline-alkali Stress. (**A**) SOD content. (**B**) POD content. (**C**) APX content. (**D**) CAT content. (**E**) $H_2O_2$ content. (**F**) DAB staining. Data showing the means $\pm$ standard deviation (SD). ($n$ = 3) (\*\* $p \leq 0.01$, \* $p \leq 0.05$).

Further, we observed $H_2O_2$ accumulation in leaves with DAB staining. A yellow product generation is due to a 3,3′-diamino-benzidine (DAB) reaction with $H_2O_2$ after saline-alkali stress. We found that the Ctrl had the deepest color, while GO + Rh plants had the lightest color, indicating the least $H_2O_2$ accumulation (Figure 5E). The $H_2O_2$ content of GO, Rh, and GO + Rh plants decreased to 26.62%, 23.45%, and 43.53% compared with the Ctrl (Figure 5F). Our findings indicated that the GO + Rh-treated could increase antioxidant enzyme activity, reducing the accumulation of hydrogen peroxide to decrease the harm of saline-alkali stress on soybean seedlings that were treated by saline-alkali.

*3.5. GO + Rh Responds to Soybean's Osmotic Substances System*

The REC and MDA are important factors in cell membrane damage. The accumulation of proline could also enhance resistance by regulating the osmotic pressure of cells' saline-alkali stress, leading to an increase in MDA and PRO contents and a decrease in REC. The contents of MDA in GO + Rh-treated were the lowest and decreased by 42.80%. The REC decreased by 46.73%. Further, the PRO content of GO + Rh-treated was 56.49% more than that of the Ctrl (Figure 6A–C). This result showed that GO + Rh treatment especially could ameliorate the negative impacts of saline-alkali representation. Meanwhile, their combination would significantly reduce the damage to the cell membrane induced by saline-alkali stress, and increase the positive regulation of GO and Rh in the combined nano-structure.

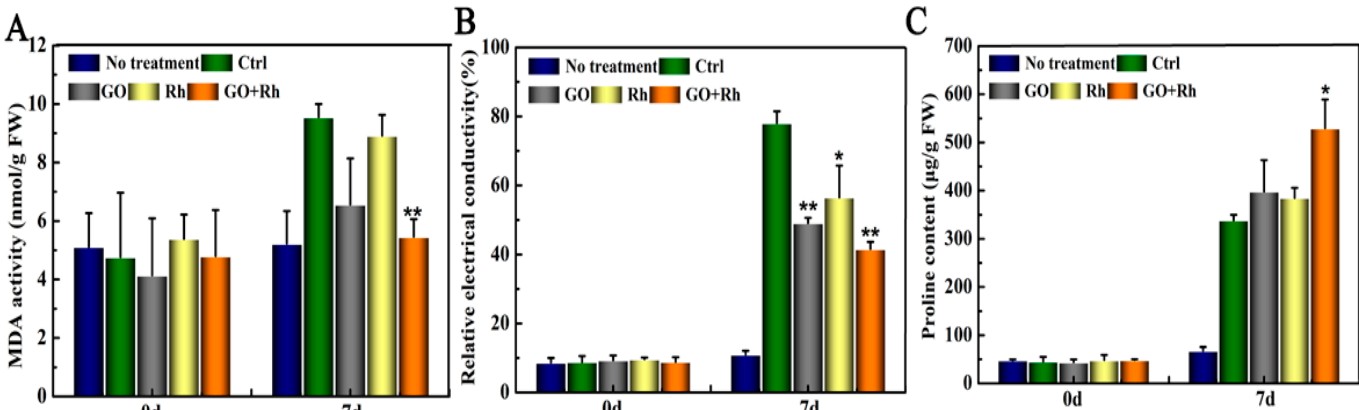

**Figure 6.** Osmotic adjustment substances content in soybean. (**A**) MDA content. (**B**) Relative electrical conductivity (REC). (**C**) Proline content (Pro). Data showing the means ± standard deviation (SD) with three replicates. * and ** represent $p \leq 0.05$ and $p \leq 0.01$, respectively.

### 3.6. GO + Rh Specifically Activates GA3 Hormones

To further study the impacts of exogenous GO, Rh, and GO + Rh on soybean hormone changes, the ABA, GA3, IAA, and JA contents were measured under saline-alkali stress. There were not many changes in ABA, GA3, IAA, and JA content in every treatment before saline-alkali stress. However, we found that the contents of ABA, GA3, IAA, and JA in the GO, Rh, and GO + Rh-treated were heightened compared with the Ctrl, respectively, and the GO + Rh-treated were significantly higher than that of the Ctrl. To be more specific, the contents of ABA, GA3, IAA, and JA in GO + Rh-treated soybeans were improved by 49.57%, 100.2%, 13.74%, and 45.51%, respectively, compared to the Ctrl under the saline-alkali stress (Figure 7D–G). Overall, our results found that the combination of GO + Rh-treated had the best effect on improving the saline-alkali tolerance of soybean by GA3 hormone increasing.

### 3.7. Relationship between Traits under Saline-Alkali Stress

The comprehensive score plot of each treatment was obtained by PCA. We know the order between the processes: GO + Rh > Rh > GO > Ctrl (Figure 8A). There are significant differences between GO + Rh-treated and other processes and GO + Rh-treated was determined to be the optimal treatment. Further, the cumulative contribution rate for PC1 and PC2 principal components is 82.1%. In detail, PC1 was severely subjected to N, $K^+$, RFW, SFW, AWE, IAA, GA3, RL, RS, RV, and RD. PC2 was strongly affected by APX, CAT, PRO, $Na^+$, SOD, POD, REC, $H_2O_2$, ABA, JA, and RW (Figure 7B).

### 3.8. GO + Rh Enhanced the Expression of Related to GA3 Hormone Genes

Four soybean's saline-alkali tolerance genes of *GmCBL1*, *GmWRI1a*, *GmGBP1*, and *GmGAMYB* related to GA3 synthesis were further selected to examine their expression levels under saline-alkali stress. We found that *GmCBL1*, *GmWRI1a*, *GmGBP1*, and *GmGAMYB* expression in GO + Rh-treated were markedly increased by 1.44, 3.59, 6.42, and 1.78 times, respectively, compared with the Ctrl (Figure 9). The findings showed these genes were up-regulated expression after saline-alkali stress. Further, the *GmGBP1* gene was the highest expression in GO + Rh-treated. To sum up, the GO + Rh treated signally enhances saline-alkali resistant gene expression related to GA3 synthesis in soybeans, thus increasing the content of GA3 and improving the plants' saline-alkali tolerance.

### 3.9. Effects of GO + Rh on N Uptake, Number of Nodules at Bloom and Pod-Setting Stage, and Yield Traits

Making full use of nodule nitrogen fixation can improve soybean quality and yield. The peak of soybean nodulation was in the period of soybean pod-setting. Our results

showed that GO + Rh-treated growth performance and yield traits were better than the Ctrl (Figure 10A,B). In detail, the pod number, empty pod numbers, seed numbers, seed weight, and 100-grain weight of GO + Rh-treated were increased by 55.81%, 72.31%, 47.51%, 58,81%, and 38.37% as compared with those of the Ctrl (Table 5). In addition, the root nodules number and N content of GO + Rh-treated were increased by 176.00% and 38.95% at the pod-setting stage (Figure 10D,E). In addition, the nitrogenase activity and leghaemoglobin content of the GO + Rh-treated soybeans increased by 39.41% and 32.15, respectively, which were positively correlated with the total nitrogen content (Figure 10F,G). We also found that the variation of $Na^+$ content in soil with different treatments was consistent with the trend in the hydroponic solution. Therefore, we speculated that the adsorption of $Na^+$ by GO + Rh reduced the over-adsorption of soybean plants. Decreased influence of $Na^+$ toxicity on soybean growth (Table 6). Our results indicated that GO + Rh could not only improve the growth development of soybean but also significantly increase the yield of soybean. It can be used as a new type of bio-compound microbial agent to improve saline alkali soil.

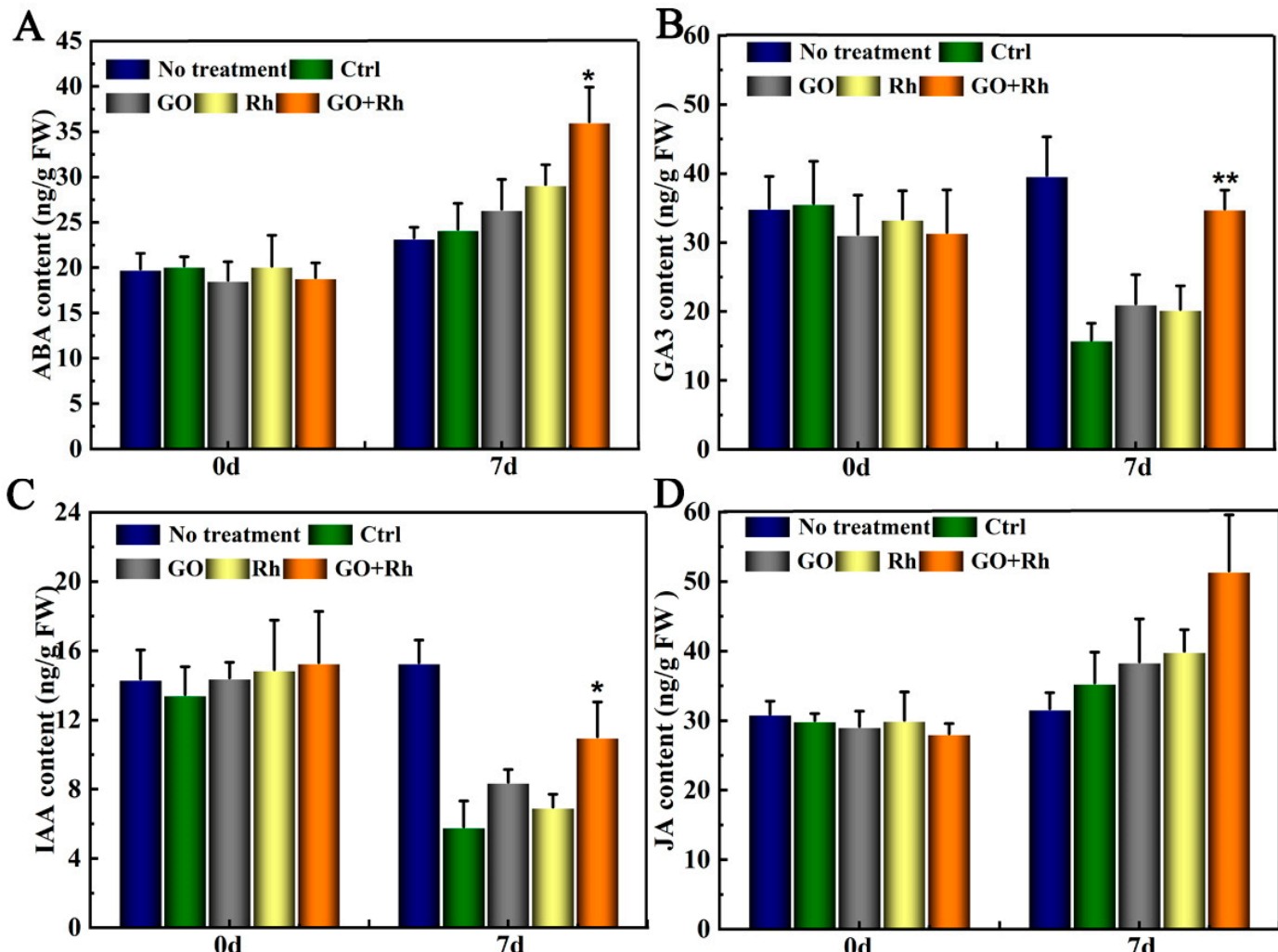

**Figure 7.** Content of saline-alkali-related hormones under saline-alkali stress. (**A**) ABA content. (**B**) GA3 content. (**C**) IAA content. (**D**) JA content. Data showing the means ± standard deviation (SD) with three replicates. * and ** represent $p \leq 0.05$ and $p \leq 0.01$, respectively.

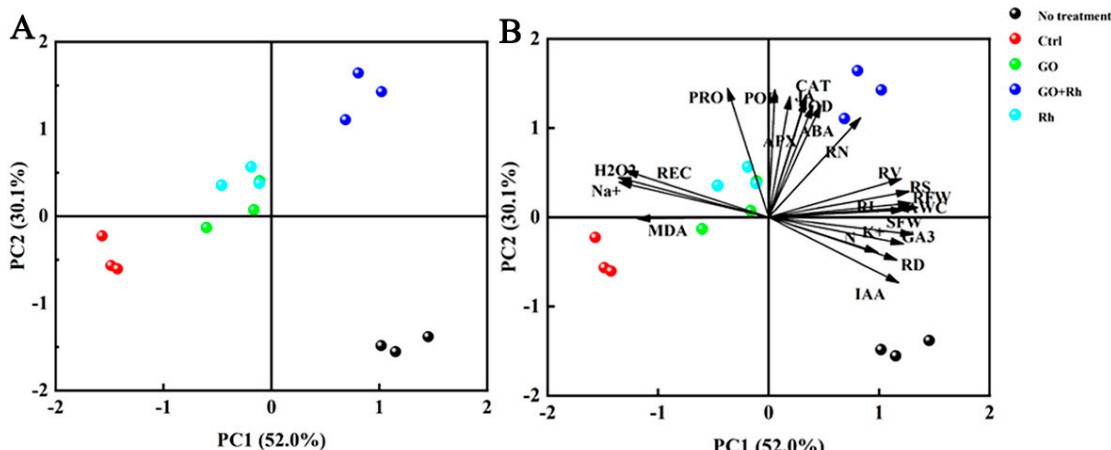

**Figure 8.** Principal component analysis (PCA) of all studied traits of soybean seedlings under saline-alkali stress. (**A**) PCA-Score plot. (**B**) PCA-Biplot. Na$^+$: Na$^+$ content; APX: APX activity; CAT: CAT activity; PRO: Proline content; SOD: SOD activity; POD: POD activity; REC: Relative electrical conductivity; MDA: MDA content; H$_2$O$_2$: H$_2$O$_2$ content; N: Nitrogen content; K$^+$: K$^+$ content; RFW: Root fresh weight; SFW: Shoot fresh weight; AWC: The absolute water of shoot; IAA: IAA content; GA3:CA3 content; ABA: ABA content; JA: JA content; RN: Root nodules; RL: Root length; RS: Root surface area; RV: Root volume; RD: Root diameter. Clockwise rotation from MDA is as follows: MDA, Na+, H2O2, REC, PRO, POD, APX, CAT, JA, SOD, ABA, RN, RV, RS, RFW, RLAWC, SFW, GA3, K+, N, RD, IAA.

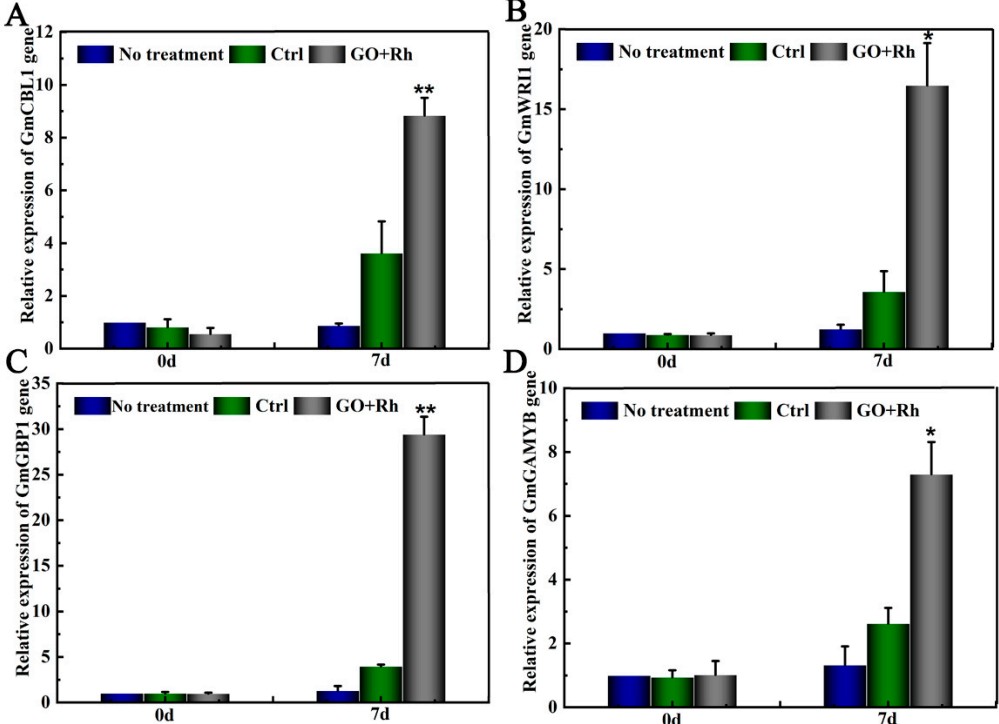

**Figure 9.** GA$_3$-related hormone gene expression in soybean. (**A**) *GmCBL*1 expression level. (**B**) *GmWRI1a* expression level. (**C**) *GmGBP*1 expression level. (**D**) *GmGAMYB* expression level. Data showing the means ± standard deviation (SD) with three replicates. * and ** represent $p \leq 0.05$ and $p \leq 0.01$, respectively.

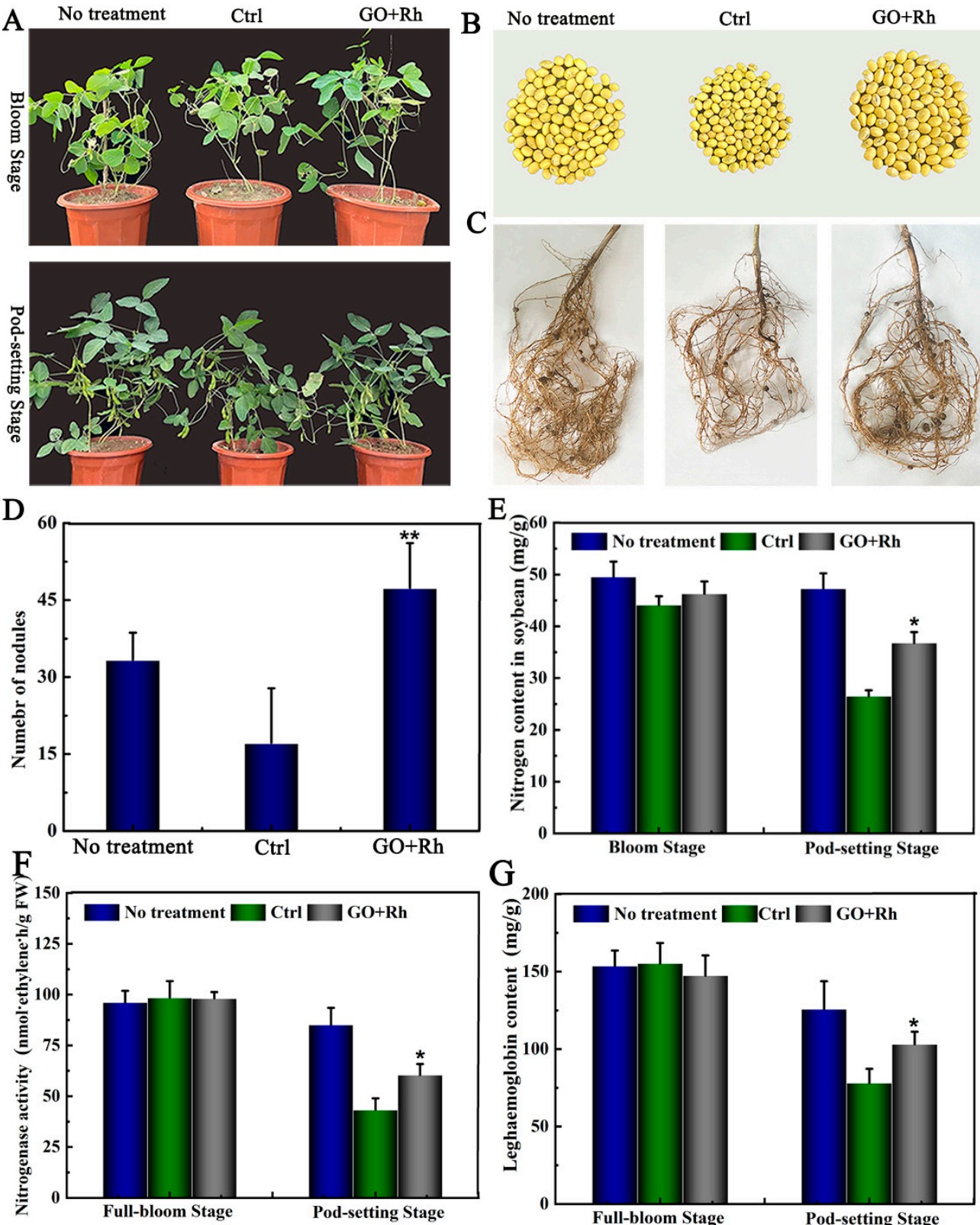

**Figure 10.** Effects of GO + Rh at bloom and pod-setting stage (**A**) Phenotype of soybean at bloom and pod-setting stage. (**B**) Pod phenotype (**C**) Root phenotype (**D**) Number of nodules. (**E**) Nitrogen content in soybean. (**F**) Nitrogenase activity (**G**) Leghaemoglobin content. Data showing the means ± standard deviation (SD) with three replicates. * and ** represent $p \leq 0.05$ and $p \leq 0.01$, respectively.

**Table 5.** Yield characters of soybean under saline-alkali stress.

| Treatment | Pods Number | Empty Pods Number | Seed Number | Seed Weight (g) | 100-Seed Weight (g) |
|---|---|---|---|---|---|
| No treatment | 16.25 ± 3.84 | 3.67 ± 1.83 | 27.92 ± 6.56 | 7.07 ± 1.89 | 30.44 ± 1.4 |
| Ctrl | 10.75 ± 3.62 | 2.42 ± 1.00 | 18.25 ± 6.22 | 3.89 ± 1.84 | 20.9 ± 1.76 |
| GO + Rh | 16.75 ± 6.59 ** | 4.17 ± 1.27 ** | 26.92 ± 12.63 * | 6.17 ± 3.37 * | 28.92 ± 3.04 * |

Data showing the means ± standard deviation (SD). ($n$ = 3) (** $p \leq 0.01$,* $p \leq 0.05$).

**Table 6.** Content of $Na^+$ in soil.

| Time | Treatment | $Na^+$ Content (mg/g) |
|---|---|---|
| Seedings period | No treatment | 1.98 ± 0.42 |
| | Ctrl | 1.66 ± 0.42 |
| | GO + Rh | 1.58 ± 0.28 |
| Harvest period | No treatment | 2.58 ± 1.01 |
| | Ctrl | 149.48 ± 1.97 |
| | GO + Rh | 165.7 ± 6.4 * |

Data showing the means ± standard deviation (SD). ($n$ = 3) (* $p \leq 0.05$).

## 4. Discussion

Excessive saline-alkali pressure will destroy the internal and external osmotic balance, produce ion toxicity to plants, affect the normal ability to absorb and transport nutrients, and even lead to the collapse of the antioxidant system, causing changes in the expression of hormones and related genes [24], which finally leads to plant growth inhibition and even death [25]. Our study also confirmed that exogenous application of GO and Rh could reduce the negative effects of saline-alkali stress on soybean, and significantly increase the fresh weight and water content of soybean plants under saline-alkali stress. It might be due to the fact that GO had hydrophilic functional groups that made it better able to prevent water evaporation. Further, Rh could promote the growth and nodulation of the root system, which helped the root system to better absorb and transport water and nutrition, and ensure the growth of the plant.

A strong root system plays an important role in the whole growth period of soybean [26]. While huge studies have shown that saline-alkali stress has damaged the root system, leading to a delay in the growth of nodules, reducing the nodulation ability of roots, and accelerating the senescence of nodules [27,28]. In our study, the root system of soybean grew slowly under saline-alkali stress, and the root nodule number decreased. Further, it decreased the nitrogen fixation ability in the soybean. The application of GO and Rh could alleviate the damage caused by saline-alkali to the root system, and ensure to absorb the nutrients. This might be because the inoculation of Rh directly promoted nodulation numbers, and then improved the nitrogen fixation ability. Moreover, the exogenous application of GO could improve water absorption capacity, and accelerate soybean growth, which has also been reported by Zhao et al. [29]. Our research findings showed that 30 mmg/mL GO promoted the total root length, root surface area, root diameter, and root volume, and inoculation with Rh promoted the nodulation, thus promoting the nitrogen fixation ability. Interestingly, the strongest root system with GO + Rh-treated was found which was consistent with previous studies that rhizobia could alleviate the salt stress of leguminous plants [30]. In addition, the total nitrogen content of soybean is positively correlated with the leghaemoglobin content of soybean and the activity of nitrogenase [23]. This positive correlation phenomenon was also found in our research. We speculate that GO might play a positive role in Rh-promoting nodulation. The findings were also reflected in the mature stage of soybean performance.

Saline-alkali stress mainly causes the production of ROS and the accumulation of osmotic substances in plant cells, which eventually leads to oxidative stress and damages the function of the plant cell membrane [31]. Plants resist oxidative stress and remove excessive ROS through their defense mechanisms. In detail, SOD is an important enzyme

for plants to resist the toxicity of ROS, which converts superoxide radicals ($O^{2-}$) into oxygen ($O_2$) and $H_2O_2$. Excess $H_2O_2$ is further converted into nontoxic water by a reaction of CAT, APX, and POD [32–34]. Our findings showed that $H_2O_2$, MDA, and PRO content in soybean increased under saline-alkali stress. While GO + Rh-treated increased the PRO content, decreased the $H_2O_2$ and MDA contents, and alleviated the oxidative damage. Moreover, the activities of SOD and POD treated with GO + Rh-treated also increased significantly, which was consistent with the previous research results [35]. Therefore, we speculated that GO + Rh-treated could regulate the antioxidant enzyme activities of plants themselves and cope with the damage caused by saline-alkali stress by eliminating excessive ROS. More importantly, the combined use of GO + Rh made the activity of antioxidant enzymes increase the most, the ability to scavenge ROS was the strongest, and the ability to resist saline-alkali stress was significantly better than that of a single treatment.

Plants preferentially absorb $Na^+$ under saline-alkali stress, which leads to the absorption of $K^+$ and nutrients being inhabited, the ratio of $K^+/Na^+$ being changed, and the ion balance being destroyed [36]. Studies have shown that the absorption of $K^+$ and the decrease in $Na^+$ concentration in soybean leaves are prerequisites for resisting saline-alkali stress [37]. Our data showed that GO + Rh-treated soybeans' $Na^+$ decreased, while $K^+$ increased significantly. It showed that GO + Rh-treated could inhibit the accumulation of $Na^+$ in cells by reducing the influx of $Na^+$ and maintaining the ion balance. We speculated that Rh-treated might be due to rhizobia promoting root nodulation and further root could better excrete excessive $Na^+$ and absorb more $K^+$ which makes the $K^+/Na^+$ ratio increase [38]. GO as a nano-material, has a strong adsorption ions ability, and led to reducing the $Na^+$ absorption of soybean. This also confirmed that the $Na^+$ content in the soil treated with GO + Rh was significantly higher than that of the control plants.

When plants suffer from saline-alkali stress, excepted removing the influence of reactive oxygen species through an osmotic adjustment mechanism and defense system, they could also resist adversity by regulating hormone content [39]. IAA, ABA, GA3, and JA play a key role in the process of plant resistance to saline-alkali stress [40]. In our study, the change in GA3 content in soybean plants treated with GO + Rh was the most significant. Previous studies have shown that GA could promote the elongation of hypocotyl by regulating the plant's growth under saline-alkali stress [41], Which was consistent with our findings. In addition, the GmCBL1, GmWRI1a, GmGBP1, and GmGAMYB genes expression levels, involved in GA3 synthesis, were increased under saline-alkali stress, all of them, and GmGBP1 gene expression levels increased most significantly. We speculated that GmGBP1 was involved in soybean resistance to saline-alkali stress. The GBP1 gene could enhance the ability of resisting saline-alkali has also been reported in Arabidopsis thaliana [42]. Therefore, we speculated that GO + Rh-treated might regulate endogenous GA3 synthesis by increasing the expression level of GmGBP1, and further participate in soybean resistance to saline-alkali stress.

Saline-alkali stress reduced the agronomic traits, quality, and economic yield of crops, which was the fundamental reason why the development of agricultural production was significantly affected by saline-alkali [43]. Our findings found that the pod number of each plant, grain weight, and 100-grain weight of soybean plant were obviously lower than those of soybean plants under normal conditions Negrão et al. [44]. pointed out that soybean yield under saline-alkali stress was only 70% of that under natural growth. It has been reported that inoculation of *rhizobium* could significantly increase the grain yield of wheat [45]. This is consistent with our finding that application of GO + Rh could significantly improve the yield traits of soybeans, resulting in fuller grains. In addition, the increase in the number of empty pods per plant might be related to the increase in the total number of pods in each plant [46]. The reason why the yield traits were significantly improved by applying GO + Rh under saline-alkali stress might be that the application of GO + Rh could alleviate the hazards brought by saline-alkali stress and make plants more tolerant to saline-alkali.

## 5. Conclusions

As GO, and Rh have positive effects on soybeans at different growth stages when they are used alone, the effect could be doubled when their combination in a nanostructure "GO + Rh" is applied. The findings showed that the GO + Rh application would alleviate the hazards caused by saline-alkali because it improves the antioxidant enzyme activity of soybean plants, ionic equilibrium, the content of related saline-alkali tolerance hormones, and the expression of saline-alkali tolerance genes, and ultimately, improving the yield indicators at the maturity stage of soybeans.

**Supplementary Materials:** The following supporting information can be downloaded at: https://www.mdpi.com/article/10.3390/agronomy13061637/s1, Table S1: Primers used in fluorescence quantitative PCR experiments.

**Author Contributions:** Experimental design, J.L. and D.B.; methodology, J.L., D.B., X.F., J.C. and J.C.; software and validation, D.B., J.L. and X.F., X.G. and J.C.; formal analysis, J.L., D.B., X.F., X.G. and J.C.; investigation, J.L., X.F. and D.B.; resources, J.L. and X.F.; data analysis and editing, J.L., D.B., X.F. and X.G.; writing original draft, X.F.; review and editing, J.L., D.B., X.F., X.G. and J.C.; supervision, J.L. and D.B.; project administration, J.L. and D.B.; funding acquisition, J.L. and D.B. All authors have read and agreed to the published version of the manuscript.

**Funding:** Postgraduate Innovation Funding Program of Hebei University (HBU2023SS014), Hebei Province Introduction of Overseas Students Funding Projects (C20220357).

**Data Availability Statement:** Data are contained within the article.

**Conflicts of Interest:** The authors declare no conflict of interest.

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
