# Peer review of "Combination of Graphene Oxide and Rhizobium Improved Soybean Tolerance in Saline-Alkali Stress"

_agronomy, doi:10.3390/agronomy13061637_

Round 1
Reviewer 1 Report
1. Please estimate concenteration of leghaemoglobin and Acetylene Reductase Assay in order to measure the nitrogenase activity.
2. In the statistics part kindly include post hoct test by performing DMRT based ranking of treatment and based on the outcomes please compare the results and re-write if required.
3. Please define which Rhizobia genus has been used to inoculate the Soyabean plants.
Author Response
Dear Editor and Reviewers,
We would like to express our sincere thanks to you for the constructive and positive comments. We have studied these comments carefully and have made the necessary revision accordingly. Please review the responses below we have made to reviewers’ comments for our submitted manuscript. All corrections are written in blue texts.
We would like to express our gratitude again to you for the comments on our paper. We are looking forward to hearing from you.
Yours sincerely,
Jianfeng Liu
E-mail: jianfengliu@hbu.edu.cn
Reviewer's comments to the Author:
Referee1:
- Please estimate concenteration of leghaemoglobin and Acetylene Reductase Assay in order to measure the nitrogenase activity.
Response: Thanks for your comments. The two key data of leghaemoglobin content and nitrogenase activity have been added in our revised manuscript and also showed in Figure 9F,G.
- In the statistics part kindly include post hoct test by performing DMRT based ranking of treatment and based on the outcomes please compare the results and re-write if required.
Response: Thank you for your suggestion. One sample-t test statistical method was used in all data of our manuscript , which does not include DMRT statistical method.
- Please define which Rhizobia genus has been used to inoculate the Soyabean plants.
Response: Sorry for our negligence. The Rhizobia genus is Bradyrhizobium japonicum. And its name was also added in the first paragraph of material and method.

Reviewer 2 Report
1. Original Submission
1.1 Recommendation
Major revision
2. Comments to Authors:
Ms. Ref. No.: manuscript-agronomy-2418130
Title: Combination of graphene oxide and Rhizobium affected soy-2 bean tolerance in saline-alkali stress.
Xiao-Hong Fu, Da-Hong Bian, Xu-Yang Gu, Jin-Feng Cao and Jian-Feng Liu
2.1. Overview and general recommendation
For reasons that should not be discussed in this manuscript, soybean (Glycine max L.) is the most important crop in the world. However, associated with several factors, the need to produce food in soil or with saline water becomes increasingly real. Therefore, understanding the halophyte degree of traditional cultivars and cultivation techniques to reduce saline/oxidative stress associated with these environments is essential for technological development to improve food production across the planet.
In this manuscript, the authors bring interesting results on the use of graphene oxide (GO) and Rhizobium (Rh) to mitigate the negative physiological effects of salt stress. Demonstrating very interesting results that can be applied on a large scale.
But I believe that major modifications must be made to improve the understanding of the manuscript and its application. Among them, I highlight two. A change and a suggestion.
The change is related to the statistical analysis used. First, it is not which treatment the comparison is being made with. I understand that each treatment (GO, Rh and GO+Rh) is being compared with SA (see comment on the use of SA and Ctrl in the next topic), but this is not written in Materials and Methods. If this analysis has been carried out, I request a change. The use of ANOVAs would allow the comparison between all tested treatments (No treatment, Ctrl, Go, Rh and GO+Rh). What would give more robustness to the discussion is whether there is synergism (or antagonism) in the use of GO+Rh. As for the suggestion, it would be to carry out a cost analysis of the application on a large scale. Showing whether the application of GO+Rh is economically viable or not. It is a simple analysis that can significantly increase the dissemination of the excellent results presented by the authors.
2.2 Major considerations
1. The names of the treatments used in the text are different from those used in the graphics. Please standardize.
2. As mentioned earlier, it is not clear how the statistical analysis was performed. I think it would be interesting to compare all treatments. Furthermore, I did not observe any difference between treatments on day 0 for all parameters studied. I believe that the authors could average all the values at day 0 and present these results at the beginning of the topic. This saves a lot of space in the Figures and Tables presented throughout the manuscript. Also facilitating the understanding of the work.
3. Greater enzymatic content does not necessarily mean greater adaptation to saline stress. It could be the opposite, actually. Analyzing halophyte species, adapted to saline environments, many times they present lower concentrations of antioxidant enzymes than glycophyte plants when confronted with saline stress. Producing enzyme means spending energy. Therefore, I believe that other factors may be interfering with the results presented. As (i) greater availability of energy (associated with greater nitrogen fixation, mainly in treatments with Rh) and (ii) GO and Rh have negative physiological effects on soybean, with greater production of enzymes with a response.
I do not mean to say that the hypotheses raised by the authors are wrong. But we must take into account that other factors may be at work and should be considered when discussing the results
2.3. Minor considerations
4. Lines 40-41. ROS are associated with salt stress in several ways. It is not just the accumulation of metals that induces ROS production. One of the possible effects of saline stress is the decrease in CO2 entry due to stomatal closure (which occurs in order to reduce evapotranspiration and the need for water absorption via the root system).
5. Line 94. Wouldn't it be interesting one treatment with GO and/or Rh without saline-alkaline solution? I think this treatment helps to understand the negative effects of both treatments.
6. Line 99. And the characteristics of the Rhizobium used? Did you buy it or grow it in the lab?
7. Lines 140-141. Explain better the method used in the determination of total nitrogen (I could not find the cited article and the doi does not work) and the calculation used.
8. Lines 158-159. Why were t-tests performed if you are comparing 5 treatments? Doing the analysis with t-tests, you picked up pairwise comparisons, inflating the alpha. A county heavily used ANOVA or Bonferroni corrections.
9. Lines 186-187. “K+ content of GO+Rh-treated performed best and increased 23.05% significantly.” Compared to which treatment?
10. Lines 189-191. I understand that it could be a sign. However, GO-treated and Rh-treated also showed Na+ concentration lower than the control treatment. But the concentration of Na in the hydroponic solution was not different (in relation to the control treatment). I believe that the authors can improve the discussion of these results.
11. Figure 3A and B. Indicate the treatments of each plant.
12. Line 274. The correct "word" is PCA.
Author Response
Dear Editor and Reviewers,
We would like to express our sincere thanks to you for the constructive and positive comments. We have studied these comments carefully and have made the necessary revision accordingly. Please review the responses below we have made to reviewers’ comments for our submitted manuscript. All corrections are written in blue texts.
We would like to express our gratitude again to you for the comments on our paper. We are looking forward to hearing from you.
Yours sincerely,
Jianfeng Liu
E-mail: jianfengliu@hbu.edu.cn
Reviewer's comments to the Author:
Referee: 2
- The change is related to the statistical analysis used. First, it is not which treatment the comparison is being made with. I understand that each treatment (GO, Rh and GO+Rh) is being compared with SA (see comment on the use of SA and Ctrl in the next topic), but this is not written in Materials and Methods. If this analysis has been carried out, I request a change. The use of ANOVAs would allow the comparison between all tested treatments (No treatment, Ctrl, Go, Rh and GO+Rh). What would give more robustness to the discussion is whether there is synergism (or antagonism) in the use of GO+Rh. As for the suggestion, it would be to carry out a cost analysis of the application on a large scale. Showing whether the application of GO+Rh is economically viable or not. It is a simple analysis that can significantly increase the dissemination of the excellent results presented by the authors.
Response: Very thanks for this comment. First of all, we apologize for the trouble caused by the failure to unify the processing names. Now, we have unified the names of treatments in our manuscript as “No treatment, Ctrl, GO, Rh, GO+Rh”. In addition, all treatments except “No treatment” were compared with “Ctrl” respectively, which have also modified in the material and method section. In addition,we think the possible reason might be the synergistic effect between GO and Rh. However, the mechanism of synergy needs our further study. At present, our findings only prove that the effect of GO+Rh treatment is obviously better than that of soybean treated by GO and Rh alone. All explanations have been added to the discussion.
We compared between the 100-grain weight of soybean treated with GO+Rh and the Ctrl, and the cost of adding graphene oxide and rhizobia, we preliminary estimate that it could increase by 600 dollars per hectare. We will further verify economically viable in the field and the relevant data will be published in another paper.
- The names of the treatments used in the text are different from those used in the graphics. Please standardize.
Response: Sorry for our negligence. We have unified all the different processing names in our manuscript into “No treatment, Ctrl, GO, Rh, GO+Rh”.
- As mentioned earlier, it is not clear how the statistical analysis was performed. I think it would be interesting to compare all treatments. Furthermore, I did not observe any difference between treatments on day 0 for all parameters studied. I believe that the authors could average all the values at day 0 and present these results at the beginning of the topic. This saves a lot of space in the Figures and Tables presented throughout the manuscript. Also facilitating the understanding of the work.
Response: There was no difference in 0 days to ensure the soybean plants consistency before salt-alkali stress and reduce the error. We used the statistical one sample-t test method. All treatments except “No treatment” were compared with Ctrl respectively.
- Greater enzymatic content does not necessarily mean greater adaptation to saline stress. It could be the opposite, actually. Analyzing halophyte species, adapted to saline environments, many times they present lower concentrations of antioxidant enzymes than glycophyte plants when confronted with saline stress. Producing enzyme means spending energy. Therefore, I believe that other factors may be interfering with the results presented. As (i) greater availability of energy (associated with greater nitrogen fixation, mainly in treatments with Rh) and (ii) GO and Rh have negative physiological effects on soybean, with greater production of enzymes with a response.I do not mean to say that the hypotheses raised by the authors are wrong. But we must take into account that other factors may be at work and should be considered when discussing the results.
Response: Thanks for this comment. The plant defense system is limited when it is subjected to saline-alkali stress, and we think that the increase of enzyme activity is not only due to saline-alkali stress. Based on our data, GO+Rh-treated could increase the tolerance of plants to saline-alkali stress by increasing the activity of antioxidant enzymes. In addition, too high or low concentration of GO will have an negative impact on plants growth, but our preliminary experiments have selected appropriate graphene oxide concentration is 30ug/ml which could promote the soybean growth and will not inhibit the growth of rhizobia. All of them has been explain in the discussion.
- Lines 40-41. ROS are associated with salt stress in several ways. It is not just the accumulation of metals that induces ROS production. One of the possible effects of saline stress is the decrease in CO2 entry due to stomatal closure (which occurs in order to reduce evapotranspiration and the need for water absorption via the root system).
Response: Thanks for these comments. In our manuscript, the production of ROS induced by metal ions is only one of the possible reasons, which has been modified accordingly in the discussion.
- Line 94. Wouldn't it be interesting one treatment with GO and/or Rh without saline-alkaline solution? I think this treatment helps to understand the negative effects of both treatments.
Response: Thanks for this comment. Our main purpose was to prove whether GO and Rh could promote soybean to alleviate the damage caused by saline-alkali stress, so each of our treatments was compared with Ctrl respectively, and we would like to consider the influence of all aspects in our future research.
- Line 99. And the characteristics of the Rhizobium used? Did you buy it or grow it in the lab?
Response: Thanks for these comments. Rhizobium has saline-alkali tolerance and high number of effective viable bacteria characteristics, which was purchased from Shanghai Biological Company.
- Lines 140-141. Explain better the method used in the determination of total nitrogen (I could not find the cited article and the doi does not work) and the calculation used.
Response: Sorry for our negligence. The determination method of nitrogen has been supplemented in material and methods, and the references have also been added accordingly.
- Lines 158-159. Why were t-tests performed if you are comparing 5 treatments? Doing the analysis with t-tests, you picked up pairwise comparisons, inflating the alpha. A county heavily used ANOVA or Bonferroni corrections.
Response: Thanks for this comment, Due to our expression not clear, it has caused you trouble. First of all, it has been supplemented in the material and methods. The analysis method adopted in this paper is one-sample t-test, and each of them was compared with Ctrl separately except No treatment.
- Lines 186-187. “K+ content of GO+Rh-treated performed best and increased 23.05% significantly.” Compared to which treatment?
Response: Thank you for your suggestion. Because our unclear expression has troubled you, the K+ content increased by GO+Rh-treated is compared with that of Ctrl. This has been revised in the results section.
- Lines 189-191. I understand that it could be a sign. However, GO-treated and Rh-treated also showed Na+ concentration lower than the control treatment. But the concentration of Na in the hydroponic solution was not different (in relation to the control treatment). I believe that the authors can improve the discussion of these results.
Response: Thanks for this comment. About this result, we preliminarily speculate that it might be because Na+ in aqueous solution could not be detected after being adsorbed by GO, and GO+Rh-treated has the strongest effect to prevent Na+ from entering plant roots, so it has a significant effect.
- Figure 3A and B. Indicate the treatments of each plant.
Response: Thanks for this comment. We have indicated the treatment of each of plant in Figure 3A and B.
- Line 274. The correct "word" is PCA.
Response: Sorry for our negligence. We have changed PAC to PCA.

Reviewer 3 Report
The paper entitled ‘Combination of graphene oxide and Rhizobium affected soyabean tolerance in saline-alkali stress; highlights the role of graphene oxide and Rhizobium in mitigating salinity stress in soyabean. I found this study very informative. However, I have following comments that needs to be addressed
I suggest authors to modify the tittle. For example replace affected with improved.
In the introduction section I suggest authors provide more information on how salinity affects soyabean with some case reports
On what basis author’s select graphene oxide and Rhizobium combined strategy for mitigating salinity stress.
Material and methods can be improved.
Although authors have carried out vast experiments but they failed to highlight in the discussion section. I suggest authors to improve this section.
Figure legends needs more information
There are some minor grammatical errors that needs to be corrected.
I recommend this paper for publication after addressing above comments
Author Response
Dear Editor and Reviewers,
We would like to express our sincere thanks to you for the constructive and positive comments. We have studied these comments carefully and have made the necessary revision accordingly. Please review the responses below we have made to reviewers’ comments for our submitted manuscript. All corrections are written in blue texts.
We would like to express our gratitude again to you for the comments on our paper. We are looking forward to hearing from you.
Yours sincerely,
Jianfeng Liu
E-mail: jianfengliu@hbu.edu.cn
Reviewer's comments to the Author:
Referee: 3
- I suggest authors to modify the tittle. For example replace affected with improved.
Response: Thanks for your comments, we have changed affected into improved.
- In the introduction section I suggest authors provide more information on how salinity affects soyabean with some case reports
Response: Thanks very much, the information on how salinity affects soybean was been added in the introduction.
- On what basis author’s select graphene oxide and Rhizobium combined strategy for mitigating salinity stress.
Response: Thanks for your comments. First of all, our previous research proves that graphene oxide could promote soybean to resist drought stress, so we speculated that graphene oxide might alleviate the damage of saline-alkali stress to soybean. In addition, we selected rhizobia with the characteristics of saline-alkali tolerance and fingdings showed that rhizobia play an important role in nodulation level and root growth, and a strong root system is necessary for plant growth. Therefore, we speculated that the combination of the GO and Rh will produce double positive effects. Therefore, the combination of GO and Rh was selected to analyze the effects on saline-alkali stress in our study.
- Material and methods can be improved.
Response: Thanks for your comments. We have modified all the materials and methods.
- Although authors have carried out vast experiments but they failed to highlight in the discussion section. I suggest authors to improve this section.
Response: Thanks for your comments. We have revised and supplemented the discussion according to the our research content and data.
- Figure legends needs more information
Response: Sorry for our negligence. The legend information and processing in the manuscript have been adjusted and supplemented.
- There are some minor grammatical errors that needs to be corrected.
Response: Sorry for our negligence. Some grammatical errors have been all corrected in our manuscript.

Round 2
Reviewer 1 Report
The manuscript can be accepted for pubication.
Reviewer 2 Report
The authors answered all questions from the first review. However, I still believe that the statistical analysis used is not correct. For example, how to say if the GO+Rh treatment is better than the GO if they are not compared to each other?
In the analysis of some graphs, it is noticed that the deviations overlap. That is, there is a strong indication that there is no statistical difference between these treatments. Contrary to the understanding of the authors, I do not see so much obviousness.
In Figure 10, the authors forgot to place the items (F) Nitrogenase activity (G) 418 Leghemoglobin content.
Author Response
Responses to the Editor and Reviewer’s comments to manuscript ( Agronomy-2418130)
Dear Editor and Reviewers,
We would like to express our sincere thanks to you for the constructive and positive comments. We have studied these comments carefully and have made the necessary revision accordingly. Please review the responses below we have made to reviewers’ comments for our submitted manuscript. All corrections are written in blue texts.
We would like to express our gratitude again to you for the comments on our paper. We are looking forward to hearing from you.
Yours sincerely,
Jianfeng Liu
E-mail: jianfengliu@hbu.edu.cn
Reviewer's comments to the Author:
Referee: 2
- The authors answered all questions from the first review. However, I still believe that the statistical analysis used is not correct. For example, how to say if the GO+Rh treatment is better than the GO if they are not compared to each other?
Response: First of all, we are very grateful to the reviewer for his (her) serious responsibility for our statistical analysis. Maybe it is because our description is not clear, which brings you confusion. The pairwise comparison was used in our statistical analysis, including GO and Ctrl, Rh and Ctrl, GO+Rh and Ctrl. In addation, the Ctrl as control, we found that GO+Rh-treated has the largest difference, which could reach a significant or extremely significant level. As other alone treatment GO or Rh, there was no significant change compared with the control, and even if there was a difference, it was not as good as the GO+Rh-treated. Moreover we could found GO+Rh>Rh>GO>Ctrl through the comprehensive score chart of PCA result. taken together, we think that GO+Rh-treated has the best effect than GO, Rh-treated alone, and not compared GO, Rh and GO+Rh-treated each other.
- In the analysis of some graphs, it is noticed that the deviations overlap. That is, there is a strong indication that there is no statistical difference between these treatments. Contrary to the understanding of the authors, I do not see so much obviousness.
Response: Thanks for this comment, In some graphs really exist the deviations overlap, but it doesn't matter, the deviations overlap are no statistical difference. Only those data that achieve significant differences are non-overlapping. we compared GO with Ctrl, Rh with Ctrl, GO+Rh with Ctrl respectively for our data and there was no deviations overlap if there was significance between them and Ctrl.
- In Figure 10, the authors forgot to place the items (F) Nitrogenase activity (G) 418 Leghemoglobin content.
Response: Thanks for these comments. Sorry for my neglect about writing Figure 10 into Figure 9 by mistake. We have added (F) Nitrogenase activity (G) Leghemoglobin content in the Figure 10.
